# Impact of FLT3–ITD Mutation Status and Its Ratio in a Cohort of 2901 Patients Undergoing Upfront Intensive Chemotherapy: A PETHEMA Registry Study

**DOI:** 10.3390/cancers14235799

**Published:** 2022-11-24

**Authors:** Rosa Ayala, Gonzalo Carreño-Tarragona, Eva Barragán, Blanca Boluda, María J. Larráyoz, María Carmen Chillón, Estrella Carrillo-Cruz, Cristina Bilbao, Joaquín Sánchez-García, Teresa Bernal, David Martinez-Cuadron, Cristina Gil, Josefina Serrano, Carlos Rodriguez-Medina, Juan Bergua, José A. Pérez-Simón, María Calbacho, Juan M. Alonso-Domínguez, Jorge Labrador, Mar Tormo, Maria Luz Amigo, Pilar Herrera-Puente, Inmaculada Rapado, Claudia Sargas, Iria Vazquez, María J. Calasanz, Teresa Gomez-Casares, Ramón García-Sanz, Miguel A. Sanz, Joaquín Martínez-López, Pau Montesinos

**Affiliations:** 1Hematology Department, Hospital Universitario 12 de Octubre, i+12, CNIO, CIBERONC, Complutense University, 28041 Madrid, Spain; 2Molecular Biology Unit, Hospital Universitari i Politécnic-IIS La Fe, 46026 Valencia, Spain; 3Hematology Department, Hospital Universitari i Politécnic-IIS La Fe, CIBERONC, 46026 Valencia, Spain; 4CIMA LAB Diagnostics, Universidad de Navarra, 31008 Pamplona, Spain; 5Hospital Universitario de Salamanca (HUS/IBSAL), CIBERONC and Center for Cancer Research-IBMCC (USAL-CSIC), 37007 Salamanca, Spain; 6Hospital Universitario Virgen del Rocío, Instituto de Biomedicina (IBIS/CSIC/CIBERONC), Universidad de Sevilla, 41120 Sevilla, Spain; 7Hospital Universitario de Gran Canaria Dr. Negrín, 35002 Las Palmas de Gran Canaria, Spain; 8IMIBIC, Hematology, Hospital Universitario Reina Sofía, UCO, 14004 Córdoba, Spain; 9Hospital Universitario Central de Asturias, Instituto de Investigación del Principado de Asturias (ISPA), 33011 Oviedo, Spain; 10Hospital General Universitario de Alicante, 03010 Alicante, Spain; 11Hospital Universitario San Pedro de Alcántara, 10001 Cáceres, Spain; 12Hospital Universitario Fundación Jiménez Díaz, 28040 Madrid, Spain; 13Research Unit, Hematology Department, Hospital Universitario de Burgos, Universidad Isabel I, 09006 Burgos, Spain; 14Hematology Department, Hospital Clínico Universitario-INCLIVA, 46026 Valencia, Spain; 15Hospital Universitario Morales Messeguer, 30008 Murcia, Spain; 16Hospital Universitario Ramón y Cajal, 28034 Madrid, Spain

**Keywords:** FLT3–ITD mutation and ratio, real-world outcomes, acute myeloid leukemia (AML), prognosis, outcome, death, relapse, survival

## Abstract

**Simple Summary:**

The prognostic impact of FLT3–ITD allele ratio (AR) is a matter of controversy. We analyzed 2901 AML patients with long-term follow-up treated with PETHEMA protocols in the pre-FLT3 inhibitors era, with 579 of them harboring the FLT3–ITD mutation. We found that FLT3–ITD AR > 0.5 was associated with lower complete remission and rate and overall survival, while AR > 0.8 was associated with lower RFS. An AR of 0.44 was the best cutoff for OS and 0.8 for RFS. Overall, allo- and auto-hematopoietic stem cell transplant (HSCT) in first CR offered similar OS in patients with AR < 0.44, while allo-HSCT improved OS for those with higher AR. However, allo-HSCT resulted in better OS and RFS as compared to auto-HSCT in NPM1/FLT3–ITD-mutated AML regardless of pre-established AR cutoff (≤0.5 vs. >0.5), supporting the use of other risk stratification tools, such as NPM1 MRD monitoring, in this setting.

**Abstract:**

FLT3–ITD results in a poor prognosis in terms of overall survival (OS) and relapse-free survival (RFS) in acute myeloid leukemia (AML). However, the prognostic usefulness of the allelic ratio (AR) to select post-remission therapy remains controversial. Our study focuses on the prognostic impact of FLT3–ITD and its ratio in a series of 2901 adult patients treated intensively in the pre-FLT3 inhibitor era and reported in the PETHEMA registry. A total of 579 of these patients (20%) harbored FLT3–ITD mutations. In multivariate analyses, patients with an FLT3–ITD allele ratio (AR) of >0.5 showed a lower complete remission (CR rate) and OS (HR 1.47, *p* = 0.009), while AR > 0.8 was associated with poorer RFS (HR 2.1; *p* < 0.001). Among NPM1/FLT3–ITD-mutated patients, median OS gradually decreased according to FLT3–ITD status and ratio (34.3 months FLT3–ITD-negative, 25.3 months up to 0.25, 14.5 months up to 0.5, and 10 months ≥ 0.5, *p* < 0.001). Post-remission allogeneic transplant (allo-HSCT) resulted in better OS and RFS as compared to auto-HSCT in NPM1/FLT3–ITD-mutated AML regardless of pre-established AR cutoff (≤0.5 vs. >0.5). Using the maximally selected log-rank statistics, we established an optimal cutoff of FLT3–ITD AR of 0.44 for OS, and 0.8 for RFS. We analyzed the OS and RFS according to FLT3–ITD status in all patients, and we found that the group of FLT3–ITD-positive patients with AR < 0.44 had similar 5-year OS after allo-HSCT or auto-HSCT (52% and 41%, respectively, *p* = 0.86), but worse RFS after auto-HSCT (*p* = 0.01). Among patients with FLT3–ITD AR > 0.44, allo-HSCT was superior to auto-HSCT in terms of OS and RFS. This study provides more evidence for a better characterization of patients with AML harboring FLT3–ITD mutations.

## 1. Introduction

FLT3–ITD mutations are present in approximately 20% of acute myeloid leukemia (AML) cases, and they are associated with leukocytosis, an increased risk of relapse, and a shorter overall survival (OS) [1,2,3,4,5,6,7]. Therefore, their presence has been consistently associated with a worse outcome in patients with AML with normal cytogenetics or intermediate-risk cytogenetics. However, the prognostic value of FLT3–ITD mutations in patients with high-risk AML is a matter of controversy, and the cutoff allelic ratio for the definition of a high or low ratio varies across different publications (from 0.5 to 1) [8].

Moreover, although the molecular mechanisms that are involved in the poor outcomes of patients who are FLT3–ITD-positive are not completely understood, several papers have shown that a high mutant allele/wild-type allele ratio (mut/wt ratio) may play a role, as those patients present poor OS and disease-free survival (DFS) [3,4,7]. The presence of a high mut/wt ratio is associated with the loss of wt FLT3, whose presence might impair the transforming potential of mutated FLT3 [7,9].

In 2014, for a series of 2278 patients with AML, the German group [10] published a different prognostication of the FLT3/ITD ratio and set the cutoff of 0.51 to define different prognoses. In FLT3–ITD-positive AML, a high allelic ratio was a prediction for low complete remission (CR) rates and poor survival. In FLT3–ITD-positive AML, an allogeneic hematopoietic stem cell transplant (allo-HSCT) in the first CR outweighed the negative impact of a high allelic ratio on survival. Moreover, this cutoff was included in the 2017 European LeukemiaNet (ELN) classification for acute myeloid leukemia.

Ravandi [11] carried out a validation of the 2017 ELN classification in 715 patients with AML and showed that the inclusion of quantitative information regarding FLT3–ITD improved risk stratification. However, risk stratification according to FLT3–ITD allele ratios is controversial and requires further evaluation in different treatment settings, such as the use of FLT3 inhibitors and allogeneic SCT.

Moreover, FLT3–ITD mutations have been associated with NPM1 mutations, whose presence seems to mitigate the effects of FLT3–ITD mutations on survival, at least in those with low mut/wt ratios [12,13]. Finally, although it is generally accepted that patients with FLT3–ITD mutations, especially those with high mut/wt ratios, must undergo an allogenic hematopoietic stem cell transplant, this has not escaped controversy, as some groups have found no differences between autografts and allografts, and one even found no differences with only chemotherapy as consolidation therapy [14,15,16,17].

To provide new insights into the characterization of FLT3–ITD mutations and their ratio, and to evaluate the prognostic value of these mutations, we set out this systematic, retrospective chart review examining real-life outcomes (patient demographics, clinical characteristics, treatment patterns, and outcomes) in Spanish and Portuguese patients with AML from the Programa Español de Tratamientos en Hematología (PETHEMA) epidemiologic registry.

## 2. Methods

### 2.1. Study Design

This was a non-interventional, systematic, retrospective chart review of data from patients enrolled in the PETHEMA registry (NCT02607059), which included patients diagnosed with AML, regardless of the treatment administered. According to the Declaration of Helsinki, informed consent was obtained from all patients, and the protocol was approved by the Research Ethics Board of each participating hospital.

### 2.2. Patients and Eligibility

Patients with de novo or secondary AML diagnosed and treated at a PETHEMA site between 1 January 1999 and 1 January 2021 were eligible for inclusion in this study. In total, 2901 adult patients enrolled in the consecutive multicenter AML PETHEMA protocols: AML 98 (N 98), AML99 (N 339), AML 07 (N 658), AML 10 (N 1336), AML 16 (N 45), and AML 17 (N 109). The remaining patients who did not follow guidelines (*n* = 316) but were treated with intensive approaches were included in this study. Patients with acute promyelocytic leukemia (i.e., M3 AML) or mixed-phenotype acute leukemia were excluded. The main characteristics of the patients are summarized in Appendix A: 1353 were female (47%) and 1544 were male (53%), the media (range) age was 73.6 (12.3–81.2), the media (range) white blood cell (WBC) count (×1000/mcl) was 38.2 (0.06–434), and regarding the cytogenetic MRC risk, 238 were low-risk (9%), 1748 were intermediate-risk (68%), and 580 were high-risk (23%). The presence of the NPM1 or FLT3–ITD mutation was detected in 826 (32%) and 579 (20%) patients, respectively.

### 2.3. Data Extraction

De-identified patient-level data from all patients meeting the inclusion criteria were entered into a secure database, with a data cutoff date of 1 January 2021. The data, which were entered into an electronic case report form, included patient demographics (i.e., age and gender), clinical characteristics (i.e., the date of diagnosis, cytogenetic status and mutation status at diagnosis and relapse, Eastern Cooperative Oncology Group (ECOG) performance status, and laboratory values at diagnosis and relapse), treatment patterns (i.e., front-line treatment regimen (type, start date, and end date), stem cell transplant (the type, date, and number of transplants)), and outcomes (i.e., response; time, date, and cause of death; and the date of the last follow-up).

### 2.4. Treatment Schedules

All patients received intensive therapy regimens, which usually included anthracycline (idarubicin (Ida) or daunorubicin (Dauno) plus Ara-C (Cyt))-based regimens: the induction treatment scheme was 3 + 7 (Cyt/Ida) in 2339 cases, 2 + 5 (Cyt/Ida) in 192 cases, 3/7 (Cyt/Dauno) in 72 cases, FLAG-I (fludarabine, Cyt, G-CSF, and Ida) in 136, ICE (Ida, Cyt, and etoposide) in 64 cases, Ida/Cyt adjusted for older patients in 50 cases, and Mitoxantrone + Cit in 48 cases. The treatment details for the first episode are detailed in Appendix A. Front-line FLT3 inhibitors (i.e., midostaurin) were not used in routine practice until 2019. For that reason, patients who received FLT3 inhibitors were not included. A total of 2851 patients received any consolidation treatment scheme. A total of 431 patients (16%) were consolidated with an automatic stem cell transplant (auto-HSCT), and 644 (23%) were consolidated with an allogeneic hematopoietic stem cell transplant (allo-HSCT) [18].

### 2.5. Cytogenetic and Molecular Analysis

Chromosome banding was performed using standard techniques (evaluated data of 2566 cases). All patients were additionally analyzed either by fluorescence in situ hybridization or by polymerase chain reaction for the presence of the recurring gene fusions *RUNX1-RUNX1T1, CBFB-MYH11,* and *PML-RARA*. Samples at diagnosis were analyzed for mutations in NPM1 (*n* = 2617) and FLT3 (ITD, N = 2901).

### 2.6. Analysis of FLT3–ITD and Ratio

Analyses of FLT3–ITD mutations were performed on bone marrow (BM) samples collected at AML diagnosis. DNA from white blood cells was obtained in each center by following previously established DNA isolation protocols. A quantitative assessment of FLT3–ITD mutations was performed using the Genescan analysis, utilizing a fluorescently labeled primer with 6-FAM, to determine allelic FLT3–ITD ratio and size, following the method described by Thiede et al. [3]. If several mutant alleles were detected by Genescan, the mutant allele with the highest allelic ratio was selected for the size analyses.

### 2.7. Definitions and Study Endpoints

The primary endpoint of the study was OS, defined as the time from the date of diagnosis to the date of death. The secondary endpoints were morphologic CR, CRi, partial remission (PR), induction death, relapse-free survival (RFS), and the frequency of allo-HSCT/auto-HSCT. The remission induction response was assessed according to the revised criteria by Cheson et al. [19]. The patient performance status at diagnosis was measured using the Eastern Cooperative Oncology Group (ECOG) scale. Cytogenetic risk groups were defined as MRC classification [20]. Measurable residual disease (MRD) negativity was defined by four-color flow cytometry (FCM) according to local protocols, as previously published [21].

### 2.8. Statistical Methods

To address the differences in CR and CRi rates among the different subsets, comparisons between categorical variables were performed using χ^2^, as well as the Wilcoxon/Mann–Whitney U test for a comparison of continuous variables. Kaplan–Meier estimates were used to calculate unadjusted time-to-event variables, and the log-rank test was used to compare them according to the different therapeutic approaches. OS was calculated from the date of the diagnosis of AML until death in all included patients. RFS was measured from the date of diagnosis until the date of PR/resistant disease, relapse from CR/CRi, or death by any cause (whichever occurred first). Univariate and multivariate logistic regression models were used to test the influence of covariates on response to induction therapy. Patients who achieved CR/CRi from 1, 2, or 3 cycles of treatment induction were defined as responders. The characteristics selected for inclusion in the multivariate analysis, using the multivariate logistic regression and Cox proportional hazards model, were those for which there was some indication of a significant association in the univariate analysis (*p* < 0.1) and, if available, those for which prior studies had suggested a possible relationship. The testing and estimation of possible cutoff values for continuous variables with respect to time-to-event end points were carried out based on maximally selected log-rank statistics [22]. All *p* values reported are 2-sided. All statistical analyses were performed using the SPSS v25 software package, IBM corp (IBM, Chicago, IL, USA).

## 3. Results

### 3.1. Pretreatment Characteristics according to FLT3–ITD Allelic Ratio (FLT3–ITD AR)

The patient characteristics of this cohort are described in Table 1. An *FLT3*–ITD mutational status was observed in 2901 patients, and 579 (20%) were ITD-positive. However, the allelic ratio was available in only 421 cases, and the median *FLT3*–ITD AR was 0.68 (range 0.1–20.6), with first and third quartiles of 0.36 and 0.90, respectively. *FLT3*–ITD-positive AML cases (*n* = 579) were associated with higher leukocytosis (median 78.8 × 10^9^/L vs. 28.3 × 10^9^/L, *p* < 0.001) and higher bone marrow blasts (median 61.3% vs. 36.8% *p* < 0.001) at diagnosis.

We separated the FLT3–ITD ARs into four increasing intervals according to a previous study to facilitate the understanding of the results. The pretreatment patient characteristics according to the distribution of FLT3–ITD ARs are shown in Table 2. Groups with higher FLT3–ITD ARs were more frequently female (*p* = 0.008), had higher leukocytosis (*p* < 0.001), and more frequently showed the co-occurrence of mutated *NPM1* (*p* < 0.001). We detected a positive correlation between WBC count and FLT3–ITD AR: rho 0.317, *p* < 0.001.

### 3.2. Response to Induction Therapy according to FLT3–ITD Allelic Ratio

Response was analyzed in 2887 (99.5%) out of 2901 patients. The complete remission (CR) rate was observed in 71.7% (*n* = 2070) of the global series and in 70.6% (*n* = 409) of the FLT3–ITD-positive group. Resistant disease (RD) was observed in 21.1% (*n* = 612) of the global series and in 18.7% (*n* = 108) of the FLT3–ITD-positive group; early death (ED) was observed in 7.9% (*n* = 229) of the global series and in 6% (*n* = 35) of the FLT3–ITD-positive group.

No differences between the levels of FLT3–ITD AR (four levels) were associated with resistance after induction treatment (Table 1). We define responders as patients who achieve CR or CRi after one or two induction cycles, and the characteristics associated with the responders group are shown in Table 2. The presence of FLT3–ITD was not associated with resistance to induction treatment. Only when comparing patients with FLT3–ITD AR > 0.5 vs. negative or ratio < 0.5 did we observe that patients with FLT3–ITD AR > 0.5 showed a lower CR rate (60.7% vs. 66.5%, *p* = 0.022). This result was reflected in the multivariable logistic regression model with the end point achievement of CR after induction therapy (OR = 0.613; *p* = 0.005) (see Table 3). Other factors associated with a lower CR rate were older age, higher WBC counts, and intermediate cytogenetic risk or high risk vs. low risk. Only the presence of NPM1 mutations was associated with the achievement of CR (OR 2.815; *p* < 0.001).

When we analyzed the MRD using FCM, we observed no differences in mean MRD, post-consolidation 1, or post-consolidation 2 between the groups who were FLT3–ITD-negative or between those with a ratio of less than 0.5 or 0.8 vs. those with a ratio greater than 0.5 or 0.8, respectively (*p* = NS in both analyses).

### 3.3. Survival Analyses according to FLT3–ITD Allelic Ratio

The media and median (range) follow-up time for survival in the 578 patients who were FLT3–ITD-positive were 3.3 years and 2.4 years (0–20.9 years), respectively.

Overall and relapse-free survival according to the FLT3–ITD allelic ratio and type of post-remission were analyzed. We confirmed the poor prognosis of a higher FLT3–ITD AR (>0.8) for OS and RFS (Appendix A). Median OS for the FLT3–ITD groups were 20.4, 18.6, 14.8, 13.5, and 11.0 months for the negative group, FLT3–ITD AR < 0.25, FLT3–ITD AR 0.25–0.5, FLT3–ITD AR 0.51–0.80, and FLT3–ITD AR > 0.8, respectively (*p* < 0.001). Median RFS median for the FLT3–ITD groups were 34.1, 32.6, 22.9, 90.7, and 11.8 months for the negative group, FLT3–ITD AR < 0.25, FLT3–ITD AR 0.25–0.5, FLT3–ITD AR 0.51–0.80, and FLT3–ITD AR > 0.8, respectively (*p* = 0.001).

In the Cox regression multivariate analyses on OS (Table 4A), male gender (HR 0.86; *p* = 0.007), older age (continuous variable, HR 1.02; *p* < 0.001), higher WBC count (continuous variable, HR 1.00; *p* < 0.001), intermediate cytogenetic risk (compared with low risk: HR 1.60; *p* < 0.001) and high cytogenetic risk (compared with low risk: HR 3.27; *p* < 0.001), FLT3–ITD AR between 0.50 and 0.80 (compared with negative: HR 1.41; 95% CI, 1.209–1.657; *p* < 0.001), and FLT3–ITD AR > 0.80 (compared with negative: HR 1.52; 95% CI, 1.248–1.853; *p* < 0.001) were revealed as being unfavorable factors for OS, whereas NPM1 mutations had no impact on the prognosis for OS and were not included in the model.

In the Cox regression multivariate analyses on RFS (Table 4B), higher WBC count (continuous variable, HR 1.00; *p* = 0.038), intermediate cytogenetic risk (compared with low risk: HR 1.74; *p* < 0.001) and high cytogenetic risk (compared with low risk: HR 2.85; *p* < 0.001), and FLT3–ITD AR > 0.80 (compared with negative: HR 2.1; *p* < 0.001) were revealed as being unfavorable factors for RFS, whereas gender, patient age, and NPM1 mutations or FLT3–ITD AR between 0.50 and 0.80 had no impact on prognosis and were not included in the model.

These multivariate results were confirmed by a selection of patients whose were not consolidated with an allogeneic transplant.

### 3.4. Impact of Low FLT3–ITD AR in Patients with Co-Occurrence of Mutated NPM1

In the NPM1-mutated subgroup, both high and low FLT3–ITD ARs had adverse prognoses for OS (Figure 1a) and RFS (Figure 1b). We analyzed the prognostic impact of low FLT3–ITD AR in patients with NPM1 mutations and confirmed that the poor prognoses of all levels of FLT3–ITD ARs were overcome by allogeneic transplantation. We adjusted these results on OS and RFS between chemotherapy (CTX)/auto-HSCT or allo-HSCT as consolidated therapy, and we detected better survival rates in the group consolidated with allogeneic transplantation for OS (Figure 1b,c) and RFS (Figure 2b,c) (*p* < 0.001 for both analyses). We found better OS and RFS rates in the group consolidated with allo-HSCT, irrespective of AR level: among FLT3–ITD AR ≤ 0.5, the median OS was 16.1 months after auto-HSCT and 53.5 months after allo-HSCT; among FLT3–ITD AR > 0.5, it was 18.7 months after auto-HSCT and 125.4 months after allo-HSCT (Appendix A).

### 3.5. Selection of Optimal Cutoff of FLT3–ITD Allelic Ratio in This Cohort

To evaluate the possible optimal cutoff points of AR with respect to time-to-event end points, we used maximally selected log-rank statistics in patients who received CTX/auto-HSCT for the clinical end points OS and RFS. We did not include those who received consolidation with allogeneic transplantation because it was shown to modify prognoses. The estimated cutoff point was 0.44 for OS (Figure 3a), and it was 0.77 for RFS (Appendix A). The 0.44 cutoff was used to perform analyses of two subgroups (high AR > 0.44 or not for OS), with the aim of evaluating the impact of allogeneic transplantation in comparison with that of auto-HSCT or CTX.

### 3.6. Impact of Consolidation Treatment according to FLT3–ITD AR in All Patients

We analyzed the OS and RFS in FLT3–ITD-positive patients, and we found that in the group of FLT3–ITD-positive patients with AR < 0.44, median OS was 39.4 months after auto-HSCT, and 88.8 months after allo-HSCT (with 5-year OS 41% and 52%, respectively, *p* = 0.86). Median RFS was shorter after auto-HSCT compared to allo-HSCT (18.7 vs. 125.7 months, respectively, *p* < 0.001). In patients with FLT3–ITD AR > 0.44, median OS and RFS was superior after allo-HSCT than after auto-HSCT. Regardless of *FLT3–ITD* AR, chemotherapy consolidation offered worse OS and RFS results (*p* < 0.001) (Figure 4).

### 3.7. Subgroup Analysis of OS and RFS by HSCT Type and Biological and Genomic Characteristics

We carried out a subgroup analysis of OS and RFS using biological and genomic characteristics (Appendix A), and no differences were observed between auto-HSCT and allo-HSCT for OS. However, for RFS, all subgroups benefitted from allo-HSCT, except for the low-risk cytogenetic group, which benefitted from auto-HSCT. For the high FLT3–ITD AR group (>0.8), no differences were observed between allo-HSCT and auto-HSCT employed as consolidation therapy.

## 4. Discussion

Our study focused on the prognostic impact of FLT3–ITD mutations and their ratio in a series of 2901 adult patients enrolled in the consecutive multicenter AML trials of the PETHEMA group in the pre-FLT3 inhibitors era, 579 of whom harbor FLT3–ITD mutations. We established an optimal cutoff of FLT3–ITD AR of 0.44 for OS and 0.8 for RFS. We show that auto-HSCT and allo-HSCT offered similar OS in patients with AR ≤ 0.44, while those with higher AR clearly benefitted from an upfront allo-HSCT. For the higher FLT3–ITD AR group (>0.8), strategies to improve allo-HSCT results are essential. Our study does not support the use of low FLT3–ITD AR to assign favorable risk category among NPM1-mutated AML, as those patients did benefit from allo-HSCT as compared to auto-HSCT. We suggest that other risk stratification tools, such as MRD monitoring, could be more efficient for implementing post-remission strategies among NPM1mut/FLT3–ITDmut patients.

Since the first description of the FLT3–ITD mutation by Nakao and colleagues [23], several groups have published works about its biological, prognostic, and therapeutic implications for patients with AML. An increased relapse rate and lower overall survival seem to be consistent amongst the different studies [1,2,3,4,5,13]. Despite this, which consolidation therapy is the best and how allele burden levels affect the prognosis of these patients remain controversial [15,16,17,24,25]. For example, we identified a new cutoff based on the maximally selected log-rank statistics obtained in this cohort of patients with AML for OS, 0.44, which is similar to the classic 0.5. Another finding in favor of the relevance of the ratio is that our multivariate analyses showed that it was an independent factor for OS (0.5 and 0.8 cutoffs) and RFS (0.8 cutoff). Given the last result, we also found that the best cutoff for RFS was 0.8 using the maximally selected log-rank statistics. However, it should be noted that prognostic classifications as ELN are mainly aimed to predict OS, and we show that 0.5 ratio is also affecting CR/CRi rate, which is a strong surrogate for OS in fit AML patients. Thus, our results could support using the 0.5 cutoff unrestricted to NPM1 AML to predict OS in the pre-FLT3 inhibitors era.

The characteristics of higher levels of FLT3–ITD ARs (0.5–0.8 or >0.8) were high WBC count, female gender, and, most frequently, the co-occurrence of the NPM1-mutated gene. No differences regarding patient age were detected. As previously described [8,13], patients who were FLT3–ITD-positive had higher WBC counts, but we also detected a correlation between WBC count and FLT3–ITD AR (rho 0.317; *p* < 0.001). The remainder of the associations found are in agreement with those found in previous studies [3,23].

All levels of FLT3–ITD ARs (four levels) influenced the outcome of patients with AML in terms of poor OS and RFS; however, we did not detect significant differences concerning CR, RD, early death, or disease relapse (Table 3a). Some authors have reported that patients harboring FLT3–ITD mutations have the same rates of CR as patients without the mutations despite their higher relapse rate and lower overall survival [3,23]. It was only when we compared patients with FLT3–ITD AR > 0.5 (vs. others) that a slightly decreased CR rate (60.7% vs. 66.5%, *p* = 0.022) was found, which is in accordance with the data of other authors [10]. In addition, in the performed multivariate regression analysis, the non-responder group was found to be made up of older patients, those with a high WBC count, those with a high cytogenetic risk, those with an FLT3–ITD AR greater than 0.5 (HR 0.617, *p* = 0.005), and those with a less frequently mutated NPM1.

Although most retrospective studies show that allogeneic transplantation is beneficial, others, with the one by the British group being the most important [24], were not able to find any differences between consolidation therapies. However, we found that allogeneic transplantation had more benefits than CTX/auto-HSCT, even in those with low FLT3–ITD AR with co-occurring NPM1 mutations (median OS was 16.7 vs. 25.3 months in the FLT3–ITD AR < 0.25 group, and median OS was 16.7 vs. 25.3 months in the FLT3–ITD AR 0.25–0.50 group, for CTX/auto-HSCT vs. allo-HSCT, respectively (*p* < 001)). This result is consistent with that previously published by Sakaguchi et al. [26], where the prognosis was unfavorable in NPM1-mut-positive AML cases with low FLT3–ITD AR when allo-HSCT was not carried out in CR1. Nevertheless, we must acknowledge that there is a bias when comparing outcomes after allo-HSCT vs. post-remission chemotherapy only, as patients dying or relapsing or with severe toxicities before transplant are included in this group. To mitigate this bias, we analyzed the impact of FLT3–ITD AR among patients receiving auto-HSCT vs. allo-HSCT, and we found that OS was similar between allo- or auto-HSCT in patients with low AR (i.e., ≤0.44), although RFS was improved after allo-HSCT. In this sense, the new European LeukemiaNet recommendations [27] for the diagnosis and management of AML in adults now considers all FLT3–ITD-mutated cases (including NPM1 cases with low FLT3–ITD ARs) within the intermediate-risk group and, therefore, candidates for allo-HSCT. Our study shows that among NPM1/FLT3–ITD-mutated patients, OS was gradually decreasing according to FLT3–ITD status and ratio (34.3 months FLT3–ITD-negative, 25.3 months up to 0.25, 14.5 months up to 0.5, and 10 months ≥ 0.5). However, allo-HSCT resulted in better OS and RFS as compared to auto-HSCT in NPM1/FLT3–ITD-mutated AML regardless of pre-established AR cutoff (≤0.5 vs. >0.5). Our findings suggest that other risk stratification tools, such as MRD monitoring, could be more efficient than AR for implementing post-remission strategies among NPM1mut/FLT3–ITDmut patients.

We observed in the multivariate Cox regression that the factors associated with death were male gender, older age, a high WBC count, a high cytogenetic risk, and FLT3–ITD AR > 0.5; however, the factors associated with relapse only included a high WBC count, a high cytogenetic risk, and FLT3–ITD AR > 0.8. The group FLT3–ITD AR O.51–0.80 seems to have a better RFS, although this was not significant. This subgroup had a better prognosis, probably influenced by allogeneic transplant from which AR > 0.8 group did not benefit.

The role of gender influence in acute myeloid leukemia’s biology and outcomes is something worthy of interest. Despite having a higher proportion of high AR, female gender was associated with better overall survival. This effect on survival has been previously described in AML [28], and better CR rates have been describe in females with FLT3–ITD AML [2]. Furthermore, the RATIFY trial showed no benefit of midostaurin on overall survival in the female group despite improving EFS [29]. Similarly, we found no effect of gender on EFS. These gender differences deserve more research efforts which probably will rely on large population studies.

The results obtained in the analysis of RFS showed that all groups saw significant benefits from allotransplantation, except for the low cytogenetic risk and high FLT3–ITD AR groups (>0.8), where the difference was not significant. In spite of the new ELN risk stratification that considers all FLT3–ITD patients in the same risk group, we suggest that patients with low AR can still benefit from an allo-HSCT in second CR (probably the decision in this subgroup could be guided by MRD monitoring after consolidation, and modulated by the type of donor and transplant center experience). However, this approach could be revised under the current standard of care with midostaurin. In addition, we consider that an AR higher than 0.8 may constitute a very high-risk subgroup where allogeneic transplant should be complemented with pre- and post-transplant strategies in order to prevent relapse. In this sense, the Ratify clinical trial [29,30], the phase III study that led to the approval of midostaurin in combination with intensive chemotherapy, showed that the benefit of adding midostaurin to intensive treatment was the greatest for TKD and for cases with FLT3–ITD ARs > 0.7.

The limitations of this study are as follows. The treatment administered to this cohort of fit patients with AML does not include the current standard treatment of patients with FLT3–ITD-positive AML. The insertion site (juxta membrane domain vs. TKD or both) and the presence of several clones with different ITDs were not evaluated. No other co-occurring genomic alterations were analyzed. Although this is a retrospective study, this work delivers important results for the current management of fit patients with FLT3–ITD-positive AML.

## 5. Conclusions

Our study confirms the FLT3–ITD AR level as an independent prognostic factor for CR, OS (cutoff 0.5), and RFS (cutoff 0.8) in this cohort of fit patients with AML in the pre-FLT3 inhibitor era. We observed that allo-HSCT or auto-HSCT improved OS in patients with a low FLT3–ITD ratio, while allo-HSCT offered OS benefit in patients with high AR with or without the NPM1 mutation. We also show that FLT3–ITD AR could be a suboptimal risk stratification tool among NPM1-mutated AML, where other strategies such as MRD monitoring could improve post-remission tailored therapies. To the best of our knowledge, this is one of the biggest real-world studies focusing specifically on the outcomes of fit patients with FLT3–ITD-mutated AML.

## Figures and Tables

**Figure 1 cancers-14-05799-f001:**
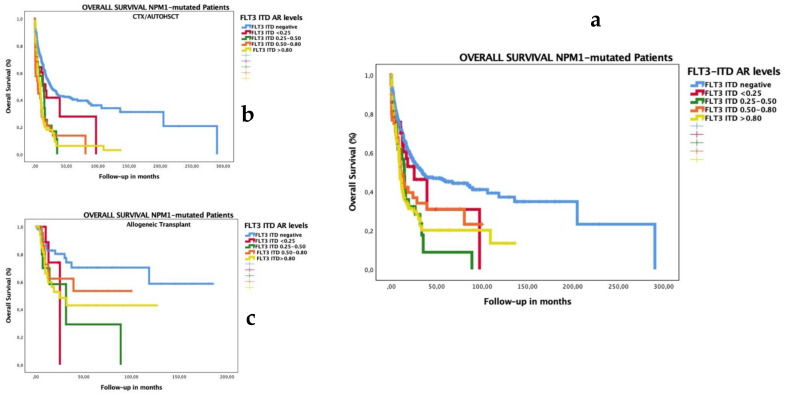
Impact on OS according to FLT3–ITD AR in patients with AML co-occurring with mutated NPM1. The OS median for NPM1-mutated with FLT3–ITD-negative group was 34.3 months (CI 14.0–54.6); with FLT3–ITD AR < 0.25, 25.3 months (8.6–42.0); with FLT3–ITD AR 0.25–0.50, 14.5 months (10.8–18.3); with FLT3–ITD AR 0.51–0.80, 10.6 months (6.4–14.9); and with FLT3–ITD AR > 0.8, 9.7 months (7.4–11.9) (*p* < 0.001) (**a**). We analyzed the impact of the test of equality of survival distribution for different levels of FLT3–ITD ARs, adjusted for consolidation treatment, CTX/auto-HSCT (**b**) or allogeneic transplantation (**c**): the OS median in the FLT3–ITD-negative group was 25.5 months vs. not reached (NR); 16.7 vs. 25.3 months in FLT3–ITD AR < 0.25; 13.9 vs. 31.7 months in FLT3–ITD AR 0.25–0.50; 4.5 months vs. NR in FLT3–ITD AR 0.51–0.80; and 8.2 vs. 24.9 months in FLT3–ITD AR > 0.80 (*p* < 001), respectively. The group consolidated with auto-HSCT or chemotherapy included 160 patients: FLT3 ITD-negative (137 cases, 68%censored), FLT3 AR < 0.25 (5 cases, 60% censored), FLT3 AR 0.25–0.50 (7 cases, 29% censored), FLT3 AR 0.501-0.80 (3 cases, 67% censored), and FLT3 AR > 0.8 (8 cases, 25% censored). The group consolidated with allo-HSCT included 143 patients: FLT3 ITD-negative (58 cases, 78% censored), FLT3 AR < 0.25 (11 cases, 73% censored), FLT3 AR 0.25–0.50 (11 cases, 45% censored), FLT3 AR 0.501–0.80 (9 cases, 65% censored), and FLT3 AR > 0.8 (37 cases, 54% censored).

**Figure 2 cancers-14-05799-f002:**
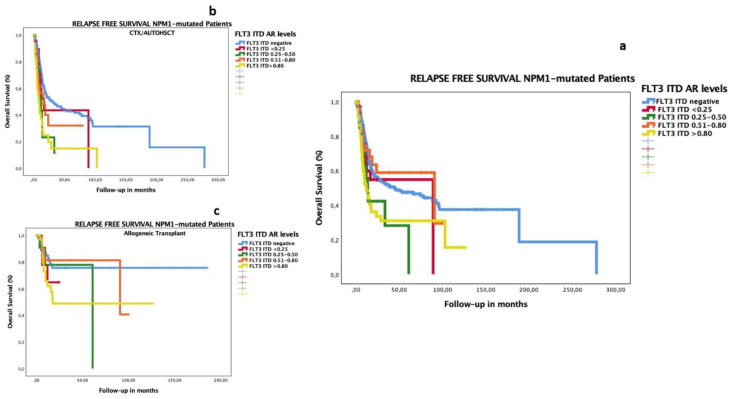
Impact on RFS according to FLT3–ITD AR in patients with AML co-occurring with mutated NPM1. The RFS median for NPM1-mutated with FLT3–ITD-negative group was 44.75 months (CI 17.4.0–72.10); that with FLT3–ITD AR <0.25 was 89.18 months (not reached); that with FLT3–ITD AR 0.25–0.50 was 13.83 months (10.9–16.7); that with FLT3–ITD AR 0.51–0.80 was 90.75 months (0–185.7); that with FLT3–ITD AR > 0.8 was 12.13 months (7.3–16.9) (*p* < 0.001) (**a**). We analyzed the impact of the test of equality of survival distribution for different levels of FLT3–ITD AR, adjusted for consolidation treatment, namely CTX/auto-HSCT (**b**) or allogeneic transplantation (**c**): the RFS median in the FLT3–ITD-negative group was 29.8 months vs. not reached (NR); 16.8 months vs. NR in FLT3–ITD AR < 0.25; 11.8 vs. 61.0 months in FLT3–ITD AR 0.25–0.50; 15.9 vs. 90.7 months in FLT3–ITD AR 0.51–0.80; and 8.4 vs. 17.8 months in FLT3–ITD AR > 0.80 (*p* < 001), respectively. The group consolidated with auto-HSCT or chemotherapy included 161 patients: FLT3 ITD-negative (138 cases, 60% censored), FLT3 AR < 0.25 (5 cases, 20% censored), FLT3 AR 0.25–0.50 (7 cases, 29% censored), FLT3 AR 0.501–0.80 (3 cases, 67% censored), and FLT3 AR > 0.8 (8cases, 25% censored). The group consolidated with allo-HSCT included 143 patients: FLT3 ITD-negative (58 cases, 78% censored), FLT3 AR < 0.25 (11 cases, 73% censored), FLT3 AR 0.25–0.50 (11 cases, 45% censored), FLT3 AR 0.501–0.80 (26 cases, 81% censored), and FLT3 AR > 0.8 (37cases, 62% censored).

**Figure 3 cancers-14-05799-f003:**
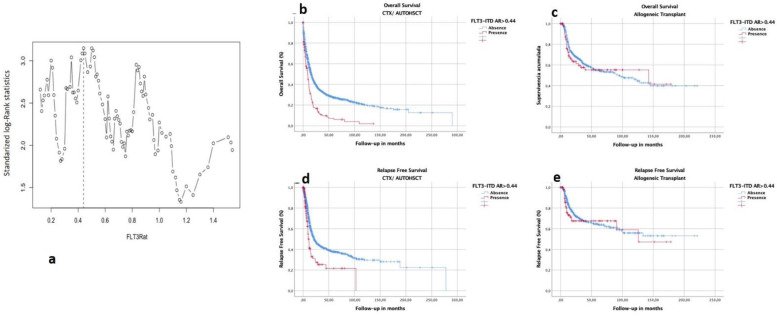
FLT3–ITD AR cutoff point with prognostic impact. Cutoff point selection using maximal log-rank statistics. Optimal cutoff point for AR by maximally selected log-rank statistics in intensively treated FLT–ITD-positive AML cases. Maximally selected log-rank statistics performed for the continuum of AR to test for a potential cutoff point by separating 2 groups with different survival distributions. AR is shown on the *x*-axis, and the corresponding standardized log-rank statistic is shown on the *y*-axis. The estimated cutoff point was 0.44, with an M statistic of 3.14 (**a**). The vertical dashed line represents the optimal cutoff point for AR evident on maximally selected log-rank statistics and corresponding M statistics. Impact on prognosis of cutoff point on OS for patients with AML consolidated with CTX/auto-HSCT (**b**) or with allo-HSCT (**c**). Impact on prognosis of cutoff point on RFS for patients with AML consolidated with CTX/auto-HSCT (**d**) or with allo-HSCT (**e**). In the groups consolidated with CTX/auto-HSCT, the estimated median OS in group with FLT3–ITD AR > 0.44 was 8.7 months (7.0–10.4) vs. 15.6 months (14.3–17.0) for group with FLT3–ITD AR< or = 0.44 (*p* < 0.001) (**b**,**d**). In the groups consolidated with allo-HSCT, the estimated median OS in the group with FLT3–ITD AR > 0.44 was 142.7 months (0–330.5) vs. 94.4 months (53.5–135.4) for the group with FLT3–ITD AR < or = 0.44 (*p* = NS) (**c**,**e**).

**Figure 4 cancers-14-05799-f004:**
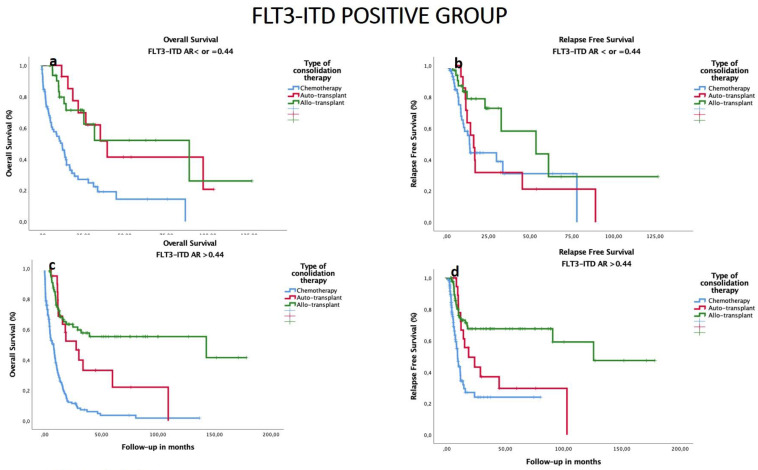
Impact of consolidation on OS and RFS in the groups with ratio of FLT3–ITD ≤ 0.44 vs. ratio > 0.44 in cohort of patients with FLT3–ITD-mutated AML. (**a**) The estimated OS median in group with FLT3–ITD AR ≤ 0.44 was 7.8 for chemotherapy, 39.4 for auto-HSCT (*n* = 15, 7 censored events), and 88.8 months for allo-HSCT (*n* = 34, 11 censored events) (*p* = 0.88 comparing auto- vs. allo-HSCT). (**b**) The estimated median RFS in group with FLT3–ITD AR ≤ 0.44 was 9.2 for chemotherapy, 18.7 for auto-HSCT, and 56.7 for allo-HSCT (*p* = 0.01). (**c**) The estimated median OS in group with FLT3–ITD AR > 0.44 was 12.3 months for chemotherapy as consolidation therapy, 27.8 for auto-HSCT (*n* = 21, 7 censored), and 59.9 for allo-HSCT (*n* = 96, 62 censored) (*p* = 0.09 comparing auto- vs. allo-HSCT). (**d**) The estimated median RFS in group with FLT3–ITD AR > 0.44 was 13.6 for chemotherapy, 16.1 for auto-HSCT, and 53.5 (6.1–100.9) for allo-HSCT (*p* = 0.01).

**Table 1 cancers-14-05799-t001:** Description of the characteristics and outcome of the patients with AML based on the levels of FLT3–ITD. A total of 2901 adult patients who enrolled in the consecutive multicenter AML PETHEMA trials and were included in the PETHEMA registry were evaluated. We separated *FLT3–ITD* ARs into 4 increasing intervals according to a previous study in order to facilitate the understanding of the results. Abbreviations: ECOG, Eastern Cooperative Oncology Group status performance; WBC, white blood cell; cytogenetic risk based on MRC classification; OS, overall survival; RFS, relapse-free survival.

Variable	FLT3 DIT NO DETECTED	FLT3 RATIO < 0.25	FLT3 RATIO 0.251–0.50	FLT3 RATIO 0.51–0.80	FLT3 RATIO > 0.80	Significance
**Age at diagnosis**						
**Mean**	53.6	53.1	53.8	53.11	52.9	*p* = NS
**Range**	12.3–81.2	19.9–76.8	17.7–75.6	17.1–77.6	13.8–81	
**Sexo**						
**Female**	1043	36	44	56	90	*p* = 0.008
**Male**	1278	39	38	48	67	
**ECOG**						
**0**	833	26	37	41	35	*p* = 0.006
**1**	838	29	25	26	70	
**2**	240	11	7	18	22	
**3**	49	1	4	5	6	
**4**	14	0	0	1	2	
**WBC count (×1000/mL)**						
**Mean**	28.28	48.86	49.46	92.5	101.06	*p* < 0.001
**Range**	0.6–340	0.7–365	0.31–298	1.7–347	1.4–434	
**Cytogenetic Risk**						
**Low Risk**	228	2	3	1	0	*p* < 0.001
**Intermediate Risk**	1314	59	61	76	130	
**High Risk**	535	5	7	9	10	
**NPM1 mutation**						
**Presence**	509	44	45	63	102	*p* < 0.001
**Absence**	1592	29	33	34	49	
**OUTCOME**	**FLT3 DIT NO** **DETECTED**	**FLT3 RATIO < 0.25**	**FLT3 RATIO 0.251–0.50**	**FLT3 RATIO 0.51–0.80**	**FLT3 RATIO > 0.80**	**Significance**
**Complete Remission, % (*n*)**	71 (1539)	69 (52)	75 (61)	55 (57)	63 (99)	*p* = 0.057
**Resistance Disease, % (*n*)**	22 (240)	20 (15)	15 (12)	29 (29)	27 (43)	*p* = NS
**Early Death, % (*n*)**	7 (64)	11 (8)	10 (8)	16 (17)	10 (15)	*p* = NS
**Relapse Disease, % (*n*)**	35 (801)	27 (20)	37 (30)	27 (28)	40 (63)	*p* = NS
**OS (median, CI), months**	20.4 (18.0–22.7)	18.6 (9.8–27.5)	14.8 (12.8–16.8)	13.5 (7.8–19.3)	11.0 (8.9–13.5)	*p* < 0.001
**RFS (median, CI), months**	34.1 (22.3–45.9)	32.7 (13.1–52.2)	22.9 (0–46.3)	90.7 (5.3–176)	32.4 (24–40.7)	*p* < 0.001

**Table 2 cancers-14-05799-t002:** Response to induction treatment. Patients who achieved CR/CRi from 1, 2, or 3 cycles of treatment induction were defined as responders. Abbreviations: WBC, white blood cell; cytogenetic risk based on MRC classification.

Variable	Responders (*n* = 1901)		NO Responders (*n* = 996)		Significance
	** *n* ** **cases**	**% of responders**	** *n* ** **cases**	**% of responders**	
**Sexo**					
Female	928	48.82	425	42.67	*p* = 0.002
Male	973	51.18	571	57.33	
**Edad**					
Mean (SD)	52.49 (13.98)		55.68 (14.36)		*p* < 0.001
**WBC count (×1000/mL)**					
Mean (SD)	35.82 (57.07)		42.89 (63.01)		*p* = 0.004
**Cytogenetic Risk**					
Low Risk	208	12.21	30	3.48	*p* < 0.001
Intermediate Risk	1226	71.95	522	60.56	
High Risk	270	15.85	310	35.96	
**NPM1 mutation**					
Presence	634	36.88	192	21.38	*p* < 0.001
Absence	1085	63.12	706	78.62	
**FLT3 ITD mutation**					
Presence	368	19.33	211	21.16	*p* = NS
Absence	1536	80.67	786	78.84	
**FLT3 ITD ratio levels**					
No mutation	1539	85.26	785	83.69	*p* = 0.057
<0.25	49	2.71	26	2.77	
0.25-0.50	61	3.38	21	2.24	
0.501-0.80	57	3.16	47	5.01	
>0.80	99	5.48	59	6.29	
**FLT3 ITD ratio > 0.5**					
Presence	164	9.09	109	11.62	*p* = 0.035
Absence	1641	90.91	829	88.38	
**FLT3 ITD ratio > 0.8**					
Presence	99	5.48	59	6.29	*p* = NS
Absence	1706	94.52	879	93.71	

**Table 3 cancers-14-05799-t003:** Factors associated with response to induction therapy. Multivariate regression logistic for response to induction treatment. Effect of patient and disease characteristics on best response to treatment (complete remissions) and multivariate analyses (prognostic factors with *p* < 0.1 in univariate analysis were included).

Variable	OR	Significance	Lower CI	Upper CI
Age	0.980	*p* < 0.001	0.973	0.987
WBC (×1000/mL)	0.996	*p* < 0.001	0.994	0.998
Cytogenetic risk				
Low risk vs. intermediate risk	0.341	*p* < 0.001	0.222	0.523
Low risk vs. High risk	0.145	*p* < 0.001	0.093	0.226
NPM1 mutation				
Absence vs. presence	2.865	*p* < 0.001	2.235	3.674
Ratio FLT3–ITD > 0.5				
Absence vs. presence	0.617	*p* = 0.005	0.441	0.862

**Table 4 cancers-14-05799-t004:** Factors associated with death and leukemia relapse. Cox multivariate for OS and RFS. Effect of patient and disease characteristics on overall survival (OS) and relapse-free survival (RFS), and multivariate analyses (prognostic factors with *p* < 0.1 in univariate analysis were included).

**A. Factors associated with death. Cox multivariate for OS**				
**Variable**	**HR**	**Significance**	**Lower CI**	**Upper CI**
**Gender (male vs. female)**	0.860	*p* = 0.007	0.772	0.960
**Age (continuous variable)**	1.020	*p* < 0.001	1.015	1.024
**WBC (×1000/mL) (continuous variable)**	1.002	*p* < 0.001	1.001	1.003
**Cytogenetic risk**				
Low risk vs. intermediate risk	1.596	*p* < 0.001	1.264	2.016
Low risk vs. high risk	3.267	*p* < 0.001	2.558	4.172
**FLT3–ITD ratio levels**				
Neg. vs. <0.25	1.404	*p* = NS	0.983	2.005
Neg. vs. 0.25–0.50	1.190	*p* = NS	0.866	1.634
Neg. vs. 0.51–0.80	1.475	*p* = 0.009	1.104	1.972
Neg. vs. >0.80	1.644	*p* < 0.001	1.305	2.072
**Consolidation (no transplant; autotransplant; allogeneic transplant)**				
No transplant vs. autotransplant	0.372	*p* < 0.001	0.311	0.445
No transplant vs. allogeneic transplant	0.321	*p* < 0.001	0.273	0.377
**B. Factors associated with relapse. Cox multivariate for RFS.**				
**Variable**	**HR**	**Significance**	**Lower CI**	**Upper CI**
**WBC (×1000/mL) (continuous variable)**	1.001	*p* = 0.038	1.000	1.003
**Cytogenetic risk**				
Low risk vs. intermediate risk	1.740	*p* < 0.001	1.331	2.275
Low risk vs. high risk	2.847	*p* < 0.001	2.118	3.826
**FLT3–ITD ratio levels**				
Neg. vs. <0.25	1.143	*p* = NS	0.713	1.833
Neg. vs. 0.25–0.50	1.366	*p* = NS	0.921	2.027
Neg. vs. 0.501–0.80	0.969	*p* = NS	0.628	1.495
Neg. vs. >0.80	2.104	*p* < 0.001	1.562	2.833
**Consolidation (no transplant; autotransplant; allogeneic transplant)**				
No transplant vs. autotransplant	0.589	*p* < 0.001	0.491	0.706
No transplant vs. allogeneic transplant	0.291	*p* < 0.001	0.239	0.354

## Data Availability

Data are property of the PETHEMA foundation. To request data, please contact Pau Montesinos (montesinos_pau@gva.es).

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
