# Peer review of "Impact of FLT3–ITD Mutation Status and Its Ratio in a Cohort of 2901 Patients Undergoing Upfront Intensive Chemotherapy: A PETHEMA Registry Study"

_cancers, 2022, doi:10.3390/cancers14235799_

Round 1

Reviewer 1 Report

In this manuscript, the authors provided a retrospective survey on the impact of FLT3ITD mutation on the prognosis of AML patients in the pre-TKI era. Analyzed patients were included in PETHEMA trials. The paper is well-written, the experimental design is appropriate and adequate and the drawn conclusion are in line with the presented results. The study has certainly some relevant limitations (correctly highligthed by the authors in the Discussion session). However, I think it has important points of strenghts, which mainly rely on the big study population. 

Some issues may be better clarified by the authors and include:

1) FLT3ITD epidemiology: it seems that a gender factor may be in place in this group of patients. To my knowledge, this point has not been deeply investigated yet. A similar point has been shown regarding the risk of death, which is increased in the male population. I would add a comment on that, maybe extending the analysis to previous studies, including RATIFY and the impact of new drugs, such as TKIs in different gender subsets. 

2) NPM1 mutated group: the results from this subses support the ELN2022 risk classification. However, it would be interesting to have some more data regarding this subset. In particular, do the authors have data regarding MRD evaluation after induction and consolidation chemotherapy as evaluated by RT-PCR? In addition, how many patients in this subgroup, after achieving CR, relapse and then undergo alloSCT in second CR? In that, what is the outcome of these patients? In general, a better characterization of this group may be relevant especially in the new ELN2002 scenario

3) High allelic ratio: although the new ELN classification removes the distinction of high vs low AR, the data that are presented in this paper (in pre-FLTi era) seem to point out that a very high risk group with an AR over 0.8 still exists. It is true that FLT inhibitors have certainly impacted on this group, but a comment on that may be provided. In particular, the role of MRD in this subset as a guide for additional approaches even after alloSCT may be discussed. In that, it would be interesting to have the results of MRD level before transplant and its impact on the outcome. 

Minor issues:

please check some typos and English gframma spelling. In some table, Spanish terms are used. 

Reviewer 2 Report

Summary:

The prognostic and predictive impact of FLT3-ITD with different allele ratio (AR) plus concurrent NPM1 mutation are still debating. In the present study, the authors studied 2901 AML patients in the PETHEMA, with 579 of them harboring the FLT3-ITD mutation. The authors found low AR and NPM1 mutation did not associated with favorable survival, and AR of 0.44 was defined as the best cut-off for prognostic indication. Moreover, the authors found that Allo-HSCT significantly improved the prognoses for patients irrespective of the FLT3-ITD AR in the NPM1-mutated group.

Comments:

1.     In Table 1, typo in the column name, “FLT3 DIT ” should be “FLT3 ITD”? Results section 204, should be Table 1?

2.     In Table 1, why RFS in AR group 0.51-0.8 was much higher than other groups? In Table 2, this group showed lower % responder than other groups; But in Table S2, this group showed higher response even compared to FLT3-ITD negative patients (HR=1.73)? In result section 250 or discussion section 378, the authors need to discuss more, e.g, whether other covariates affected the result, association of AR with other characteristics like blast count, cytogenetic risk, consolidation treatment, etc.

3.     Table 2 should also include % responder in each specific group, e.g., % responder in patients with and without FLT3-ITD, etc. The current format is not clear enough to make the comparison. And the patient numbers are confusing, e.g., the total patient# by adding AR<0.25 and 0.25-0.5 aren’t the same in the underearth group absence of AR>0.5.

4.     For all the Kaplan-Meier survival curves, patient numbers and p-values should be indicated for better interpretation of the results. Figure 1 and 2, the plots were not clear enough, couldn’t see the censored events. For example, in 1c, hard to tell why the curve of AR<0.25 dropped from >0.7 to almost 0, unless the patient number was very small.

5.     Figure 1 and 2, and discussion section 366, it wasn’t clear whether NPM1 mutation affects the survival within the low AR group (<0.25 or 0.44); One additional analysis needs to be done to verify: OS/PFS comparison in NPM1 mutated vs WT patients within the low AR group.

6.     Figure 3a and discussion section 376, AR 0.5 seemed to be a good cut-off as well, and it has been validated before (e.g., PMID: 25270908, 23377436, and 30341082), the present study should demonstrate the superiority of 0.44 over 0.5.
